# LabelG: Consistent Pairwise 3D CT Image and Segmentation Mask Generation via Medical Foundation Models

**Lu-Yan Wang**[1]  ⓘⅅ           NTHU2872@GAPP.NTHU.EDU.TW
[1] *Department of Computer Science, National Tsing Hua University, Taiwan*
**Tzung-Dau Wang**[2]            TDWANG@NTU.EDU.TW
[2] *Cardiovascular Center and Division of Cardiology, Department of Internal Medicine, National Taiwan University Hospital, Taiwan*
**Shang-Hong Lai**[1]  ⓘⅅ           LAI@CS.NTHU.EDU.TW

**Editors:** Accepted for publication at MIDL 2026

## Abstract

Medical image generation is increasingly used for data augmentation in tasks such as segmentation. However, most existing approaches focus solely on synthesizing high-quality images, while the corresponding segmentation masks are generated separately or may lack structural alignment with the images. To address this limitation, we introduce LabelG, a lightweight module that works with pretrained 3D CT diffusion foundation models to produce paired CT images and segmentation masks in a single sampling pass. LabelG decodes multi-scale latent features using a split-MLP architecture and aggregates predictions via a voting mechanism to yield anatomically coherent image–mask pairs, without requiring ground-truth masks or textual prompts at inference time. Experiments on four CT datasets demonstrate that the generated pairs achieve high visual fidelity and can improve downstream segmentation performance when used to augment limited real data. LabelG offers an efficient and scalable approach for synthesizing paired medical data, helping enhance data efficiency in medical image segmentation.

**Keywords:** Medical image generation, Segmentation mask generation, Medical AI.

## 1. Introduction

Learning downstream tasks in medical imaging plays a crucial role in assisting clinical diagnosis. For instance, medical segmentation models can help doctors quickly understand a patient's condition, while classification models can aid in preliminary diagnoses, thereby reducing the workload of medical professionals. Due to the limited availability of medical imaging datasets, an increasing number of studies have focused on generating synthetic medical images to enhance the performance of downstream tasks. DiffTumor(Chen et al., 2024) seeks to enhance the robustness and generalizability of tumor segmentation models across various organs, it can generate high-quality synthetic tumors medical images using existing tumor masks. However, its reliance on masks with pronounced boundaries not only compromises the realism of the generated images but also limits mask diversity.

To address this issue, our work introduces a segmentation module that seamlessly integrates with CT generation models for joint generation of paired CT images and segmentation masks based on a CT foundation model. By incorporating an innovative mask generation strategy with a cross-domain consistency constraint, we ensure anatomical coherence between images and masks. Our framework is trained in two stages: (1) fine-tuning

a diffusion-based CT foundation model; (2) freezing the generator and training a mask generation branch on its latent features. At inference time, a single latent code produces both an image and its mask in one forward pass. In summary, our key contributions are listed below:

- We propose a CT foundation model–based framework for single-pass generation of paired 3D CT images and masks.

- We introduce a multi-scale VAE feature mask generator with split-MLP voting for consistent, memory-efficient mask synthesis.

- Comprehensive experiments on four CT datasets showing the effectiveness of our proposed method in both generative quality and downstream segmentation.

## 2. Related Work

### 2.1. Medical Image Generation

Recent advances in deep generative modeling have been driven primarily by two families of models: generative adversarial networks (GANs) (Goodfellow et al., 2014) and diffusion models (Ho et al., 2020). GAN-based methods first popularized conditional generation for vision tasks, including image-to-image translation with models such as Pix2Pix (Henry et al., 2021) and Pix2PixHD (Wang et al., 2018). These ideas were later adapted to 3D medical imaging with volumetric GANs such as HA-GAN (Sun et al., 2022), but training instability and mode collapse remain common issues in practice.

Motivated by the higher fidelity and more stable optimization of diffusion models, a growing line of work has adopted conditional diffusion frameworks for 3D medical image synthesis. Examples include efficiency-oriented designs like 3D MedDiffusion (Wang et al., 2024) and Make-A-Volume (Zhu et al., 2023), as well as methods exploring diverse conditioning signals, such as Med-DDPM (Dorjsembe et al., 2024). Among large-scale systems, MAISI (Guo et al., 2025) stands out by adapting a ControlNet-like architecture (Zhang et al., 2023b) to enable detailed, mask-based conditioning. In parallel, text-driven systems like GenerateCT (Hamamci et al., 2023) synthesize CT volumes from free-form medical prompts. However, most of these approaches are primarily designed for image quality and flexibility in conditioning, and often provide only limited support for downstream tasks such as segmentation, where explicit and structurally consistent **image-mask pairs are required.**

### 2.2. Pair Medical Datasets Generation

Due to the high time and labor cost of manual annotation, recent works have explored paired dataset generation to expand labeled data for downstream tasks. DatasetGAN (Zhang et al., 2021) leverages pretrained GAN generators and learns a decoder that maps latent codes to pixel-wise semantic labels, enabling dense annotations from only a small set of manually labeled examples.

In the medical domain, most paired data generation methods are diffusion-based. Sim-Gen (Bhat et al., 2025) directly generates surgical images and segmentation masks from

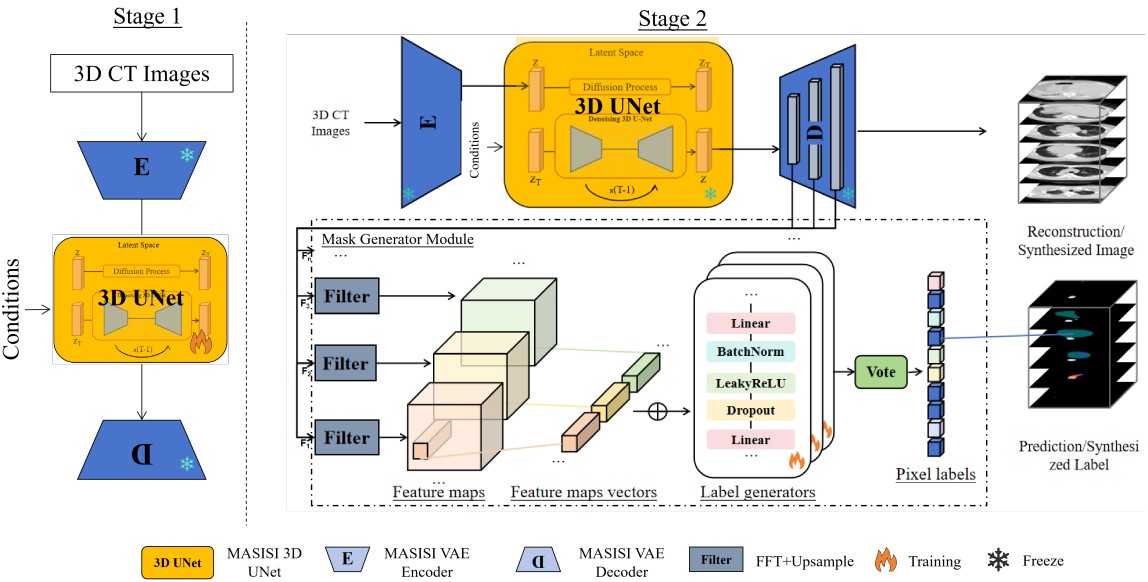

Figure 1: LabelG consists of two main components: (1) Fine-tuning the CT image foundation generation model to enhance the accuracy of body region synthesis, and (2) A segmentation mask generation branch to produce high-quality paired datasets for downstream segmentation tasks.

noise, while MedSegFactory (Mao et al., 2025) employs a dual-stream diffusion architecture to jointly synthesize images and masks with tight cross-attention coupling. However, these approaches are largely designed for the 2D image domain and are not tailored to 3D CT foundation models. In contrast, our method targets 3D CT and reuses a pretrained CT diffusion backbone, learning to generate both images and masks jointly from a shared latent representation without requiring masks or text prompts as input conditions at sampling time.

## 3. Methodology

Our framework consists of two main stages(see Fig. 1). In the first stage, we fine-tune a CT foundation model on new datasets to adapt to varying CT machine parameters, enhancing the quality of generated CT volumes. In the second stage, the fine-tuned 3D U-Net and the VAE encoder-decoder is kept frozen. We only optimize the mask generator branch on top of the denoised latent feature. We extract features from the denoised images using a frozen VAE decoder and input them into our mask generation module to produce the corresponding segmentation masks.

### 3.1. Madical Image Generation Foundation Model

MAISI(Guo et al., 2025) is a foundational generative model for medical imaging that generates CT images conditioned on segmentation masks using ControlNet(Zhang et al.,

2023a). However, this image-to-image approach relies on conditional guidance of segmentation masks, which can limit the diversity of generated images. Given its strong ability to generate high-quality 3D CT images, we first fine-tune MAISI's 3D U-Net model to enhance realism for our dataset. We first fine-tune MAISI's 3D U-Net backbone on the target CT dataset to adapt the generator to this domain and enhance realism. In the following, we keep this 3D U-Net fixed and use it as the backbone for our downstream applications. Following MAISI, we condition the diffusion model on the body-region indices and voxel spacing, and denote the primary conditions as $c_p := \{i_{\text{top}}, i_{\text{bottom}}, s\}$, which is as same as MAISI. Formally, the training objective of the diffusion model is as follows:

$$\mathbb{E}_{\mathcal{E}(x), \epsilon \sim \mathcal{N}(0,1), t, c_p} \left[ \|\epsilon - \epsilon_\theta(z_t, t, c_p)\|_1 \right] \tag{1}$$

## 3.2. A Lightweight Mask Generation Module

Built upon the pretrained CT image diffusion model, we introduce a lightweight module for generating segmentation masks alongside the synthesized CT images. Formally, let $\mathbf{z} \sim \mathcal{N}(0, I)$ be a randomly sampled latent variable, and let $\mathcal{G}(\cdot)$ denote the MAISI generative model. Given $\mathbf{z}$, the proposed module processes the multi-scale latent features of $\mathcal{G}$ and produces a corresponding segmentation mask through the following components:

### 3.2.1. Basic Generation and Feature Extraction

Given a latent variable $\mathbf{z} \sim \mathcal{N}(0, I)$ and the primary conditions $c_p$, we feed them into the frozen generator $\mathcal{G}(\cdot)$ to synthesize a 3D CT image $\mathbf{x} \in \mathbb{R}^{C \times H \times W \times D}$, where $C$ denotes the number of channels and $H, W, D$ represent the spatial dimensions. Because $\mathcal{G}$ is a pre-trained CT foundation model, this step ensures anatomically plausible outputs:

$$\mathbf{x} = \mathcal{G}(\mathbf{z}, c_p). \tag{2}$$

We then extract multi-scale feature maps $\mathbf{F} = \{\mathbf{F}_1, \mathbf{F}_2, \ldots, \mathbf{F}_n\}$ from the frozen MAISI VAE decoder $\mathcal{D}(\cdot)$. Each feature map $\mathbf{F}_i$ is upsampled to a standardized resolution $(H', W', D')$ using an interpolation function $\mathcal{U}(\cdot)$:

$$\mathbf{F}'_i = \mathcal{U}(\mathbf{F}_i), \quad i = 1, \ldots, n. \tag{3}$$

The upsampled feature maps are concatenated to form a unified feature representation:

$$\mathbf{F}_{\text{concat}} = \text{concat}(\mathbf{F}'_1, \mathbf{F}'_2, \ldots, \mathbf{F}'_n). \tag{4}$$

To reduce high-frequency artifacts in the latent features, we apply a simple FFT-based low-pass filtering to $\mathbf{F}_{\text{concat}}$ before feeding it into the segmentation heads. Concretely, we compute the 3D discrete Fourier transform (FFT) of $\mathbf{F}_{\text{concat}}$, attenuate coefficients above a fixed frequency cutoff, and then transform the result back to the spatial domain. This lightweight filtering empirically reduces background noise while preserving the main anatomical structures. In addition, we crop the filtered feature map as $\mathbf{F}_{\text{cropped}}$ to the primary target region according to the dataset-specific field of view, so that the subsequent segmentation module focuses on the most relevant anatomical structures.

### 3.2.2. Feature Vectorization and Segmentation Model Training

To render the cropped feature maps compatible with lightweight segmentation heads, we first vectorize them into a feature representation $\mathbf{v} \in \mathbb{R}^d$, where $d$ is the feature dimension:

$$\mathbf{v} = \mathcal{V}(\mathbf{F}_{\text{cropped}}). \tag{5}$$

Given the high dimensionality of $\mathbf{v}$, we partition its channels into $m$ disjoint groups, denoted as $\{\mathbf{v}_1, \ldots, \mathbf{v}_m\}$, and attach an independent prediction head $\mathcal{H}_i(\cdot)$ to each group:

$$\hat{\mathbf{y}}_i = \mathcal{H}_i(\mathbf{v}_i), \quad i = 1, \ldots, m. \tag{6}$$

The split-MLP design avoids the prohibitive memory usage of a large 3D CNN over full concatenated features, while allowing each head to specialize on different feature subsets. Their outputs are later fused to obtain a robust final mask.

### 3.2.3. Category Decision and Objective Function

Each prediction head $\mathcal{H}_i$ outputs voxel-wise class probabilities $\hat{\mathbf{y}}_i \in \mathbb{R}^{K \times H' \times W' \times D'}$, where $K$ denotes the number of classes. To produce the final prediction, we aggregate the $m$ expert outputs through a voting mechanism $\mathcal{V}(\cdot)$:

$$\hat{\mathbf{y}} = \mathcal{V}(\hat{\mathbf{y}}_1, \hat{\mathbf{y}}_2, \ldots, \hat{\mathbf{y}}_m). \tag{7}$$

In practice, $\mathcal{V}(\cdot)$ performs class-wise majority voting across the $m$ heads. In the case of ties, the class with the highest mean probability across all heads is selected.

Given a ground-truth mask $\mathbf{y} \in \{0, \ldots, K-1\}^{H' \times W' \times D'}$, each head is trained independently using a standard 3D segmentation objective that combines voxel-wise cross-entropy and soft Dice loss. The overall loss is

$$\mathcal{L}_{\text{seg}} = \frac{1}{m} \sum_{i=1}^{m} \big( \mathcal{L}_{\text{CE}}(\hat{\mathbf{y}}_i, \mathbf{y}) + \lambda \, \mathcal{L}_{\text{Dice}}(\hat{\mathbf{y}}_i, \mathbf{y}) \big), \tag{8}$$

where $\lambda = 0.2$ in all experiments.

## 4. Dataset and Implementation Details

**Datasets.** To evaluate our approach across different anatomical regions and label granularities, we use four 3D CT datasets: the public AbdomenCT dataset (Sage Bionetworks, 2025), the SegTHOR dataset (Lambert et al., 2020), the MSD10 Colon dataset (Antonelli et al., 2022), and an internal clinical dataset with fine-grained annotations.

The AbdomenCT dataset contains 600 abdominal CT scans with segmentation maps for 13 abdominal structures. We choose this dataset to assess our method on multi-organ abdominal anatomy, which is a common target for clinical segmentation. The SegTHOR dataset comprises 40 thoracic CT scans with four labeled structures, allowing us to validate that the proposed framework generalizes from the abdomen to the chest region. To further examine the ability to model tumor lesions structures, we include the MSD10 Colon dataset, which provides 190 CT scans with two classes (colon tumor and background), focusing specifically on lesion-level segmentation. Finally, we use a private thoracic–cardiac

Table 1: Performance of the baseline model's generated image quality across four datasets, the reported values represent the mean scores computed over the three axes of the 3D medical images.

| | CVAI | | Abdomen | | SegTHOR | | MSD10 Colon | |
|---|---|---|---|---|---|---|---|---|
| | FID ↓ | LPIPS ↑ | FID ↓ | LPIPS ↑ | FID ↓ | LPIPS ↑ | FID ↓ | LPIPS ↑ |
| DDPM(Ho et al., 2020) | 35.64 | 0.33 | 30.63 | 0.373 | 28.67 | 0.36 | 29.87 | 0.38 |
| LDM(Rombach et al., 2022) | 21.41 | 0.3 | 22.73 | 0.374 | 22.58 | 0.37 | 26.27 | 0.39 |
| HA-GAN(Sun et al., 2022) | 22.04 | 0.24 | 25.03 | 0.377 | 17.65 | 0.27 | 23.51 | 0.28 |
| MAISIs(Guo et al., 2025) | 17.05 | 0.45 | 14.69 | 0.55 | 21.29 | 0.41 | 17.62 | 0.46 |
| LabelG | **11.98** | **0.55** | **11.00** | **0.56** | **10.52** | **0.54** | **11.10** | **0.62** |

CT dataset with 143 scans and five segmentation labels. The annotations focus on cardiac and aortic structures, with the aorta further subdivided into four anatomically meaningful segments. All labels are manually annotated in three dimensions and reviewed by experienced radiologists, providing fine-grained and clinically curated annotations for additional evaluation.

All 3D images are reoriented to a canonical RAS orientation, intensity-clipped to $[-1000, 1000]$, and linearly normalized to $[0, 1]$. Volumes are resampled to a reference spacing of $1 \times 1 \times 0.75$ mm using linear interpolation for images and nearest-neighbor interpolation for segmentation masks, resulting in a fixed resolution of $256 \times 256 \times 128$.

**Experimental Setting** Each segmentation expert is implemented as a four-layer MLP and trained for up to 5,000 epochs on a single NVIDIA RTX 4090 GPU (24 GB memory). We extract feature maps from 8 decoder blocks of the frozen VAE, each with stride 2. These feature maps are upsampled to a common resolution using trilinear interpolation and then concatenated before being fed into the MLPs. We use a batch size of 1 and the Adam optimizer with $\beta = (0.9, 0.999)$. The initial learning rate is set to $1 \times 10^{-4}$ and is decayed every 200 epochs. To mitigate overfitting, we apply dropout with a probability $p = 0.3$ and include L1 regularization term on the network weights.

## 5. Results and Ablation Study

**Evaluation Of Synthesis Quality** Although MAISI demonstrates impressive generation quality, our work requires fine-tuning on specific datasets to achieve more accurate body region generation. To further evaluate the effectiveness of our approach, we compare the quality of our synthetic data against four baseline models: DDPM(Ho et al., 2020), LDM(Rombach et al., 2022), HA-GAN(Sun et al., 2022), and MAISI. The evaluation is conducted using the mean Fréchet Inception Distance (FID)(Heusel et al., 2017) and Learned Perceptual Image Patch Similarity (LPIPS)(Zhang et al., 2018) across four different datasets.

Table 1 demonstrates that our work achieves the best balance between fidelity and diversity, followed by MAISIs, which benefits from fine-tuning to improve body region accuracy. Traditional generative models like DDPM and LDM show weaker performance, with higher

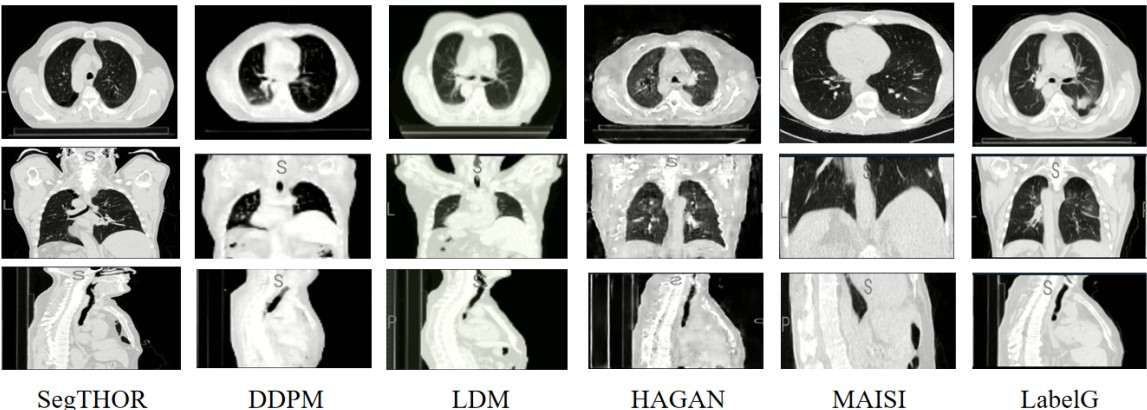

| SegTHOR | DDPM | LDM | HAGAN | MAISI | LabelG |

Figure 2: The three-axis views of the generated CT images. The left column represents the corresponding views from the original dataset, serving as a reference for comparison.

FID and lower LPIPS scores. According to the Figure 2, fine-tuning on specialized datasets enables conditionally generated body regions to better align with the original dataset.

**Subjective Realism Evaluation** To assess the perceptual realism of the generated CT images, we conducted a small-scale reader study with two radiologists. As shown in Fig. 3, our model was demonstrated as the most realistic slice with a probability of 56.3% from the 8 single-choice questions, outperforming all baseline generative models. Interestingly, the real CT slices were selected 31.3% of the time. This indicates that our model occasionally produces smoother or cleaner textures that radiologists perceive as highly realistic.

To further support this observation, we performed a binomial significance test on the single-choice results. The preference for our method is statistically significant compared to random chance ($p < 0.01$), with a 95% confidence interval of [0.33, 0.78], indicating that the observed preference is unlikely to be due to random variation.

In the multi-choice setting, real CT slices achieved the highest selection rate (52.2%). Our model ranked second with 39.1%, which is substantially higher than DPPM and HAGAN. In this setting, the difference between our method and chance selection is not statistically significant ($p > 0.05$), which we attribute to the more permissive evaluation protocol that allows simultaneous selection of real images.

In summary, the subjective evaluation demonstrates that our approach produces high-fidelity and clinically plausible CT images. We note that this reader study involves a limited number of expert readers, and a larger-scale expert evaluation remains an important direction for future work.

**Evaluation of the Benefits for Downstream Segmentation Tasks.** To further demonstrate the effectiveness of LabelG's synthetic paired datasets in improving segmentation performance, we evaluate four representative segmentation backbones—SwinUNet (Cao et al., 2022), UNETR (Hatamizadeh et al., 2022), nnUNet (Isensee et al., 2018), and SegResNet (Myronenko, 2018)—using the Sørensen–Dice coefficient (DSC) on four benchmark datasets: CVAI, Abdomen, SegTHOR, and MSD10-Colon. For all experiments, we adopt

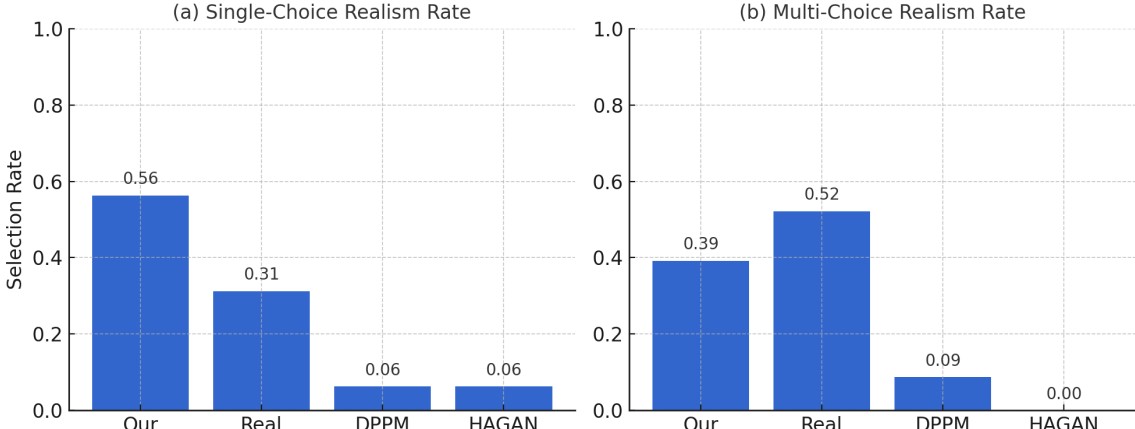

Figure 3: Radiologist assessment of image realism across two tasks. (a) Single-choice test, where radiologists selected the most realistic slice among four candidates (three generated, one real). (b) Multi-choice test, where radiologists could select two slices they considered most realistic.

Table 2: Segmentation performance (DSC) of four backbone models under different training settings: using only real data, using real data with MAISI-generated synthetic data, and using real data with LabelG-generated synthetic data.

| Dataset | Backbone | Real Only | Real + SYN(MAISI) | Real + SYN(LabelG) |
|---|---|---|---|---|
| CVAI | SwinUNet | 0.851 | 0.921 | **0.941** |
| | UNETR | 0.853 | 0.882 | **0.901** |
| | nnUNet | 0.934 | 0.921 | **0.942** |
| | SegResNet | 0.882 | 0.894 | **0.912** |
| Abdomen | SwinUNet | 0.800 | 0.790 | **0.802** |
| | UNETR | 0.801 | 0.804 | **0.831** |
| | nnUNet | 0.814 | 0.821 | **0.832** |
| | SegResNet | 0.792 | 0.801 | **0.816** |
| SegTHOR | SwinUNet | 0.816 | 0.926 | **0.948** |
| | UNETR | 0.814 | 0.824 | **0.842** |
| | nnUNet | 0.901 | 0.926 | **0.932** |
| | SegResNet | 0.802 | 0.813 | **0.821** |
| MSD10-Colon | SwinUNet | 0.423 | 0.427 | **0.429** |
| | UNETR | 0.415 | 0.420 | **0.421** |
| | nnUNet | 0.425 | **0.441** | 0.434 |
| | SegResNet | 0.412 | 0.413 | **0.414** |

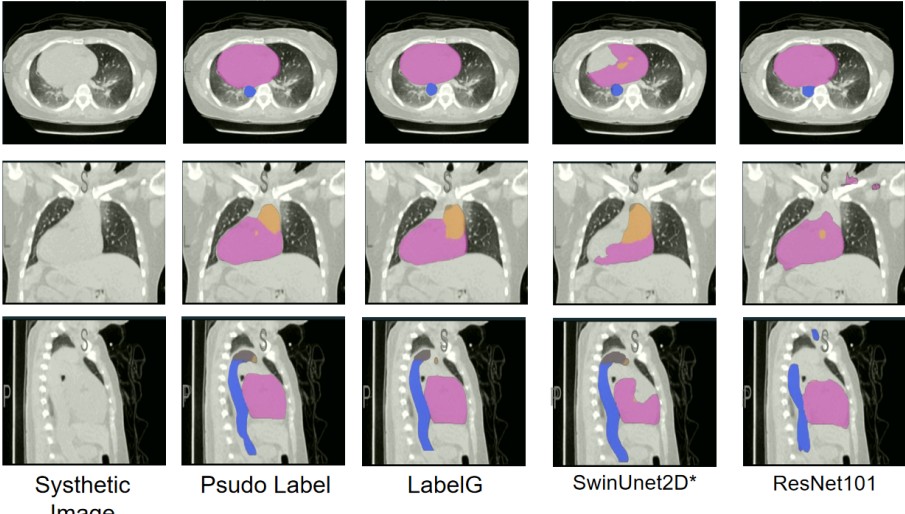

|   |   |   |   |   |
|:-:|:-:|:-:|:-:|:-:|
| Systhetic Image | Psudo Label | LabelG | SwinUnet2D* | ResNet101 |

Figure 4: Qualitative examples of generated images and their corresponding pseudo labels used in the backbone ablation study. A SwinUNet model trained on the original CVAI dataset is used to produce the pseudo labels. The asterisk (*) marks results produced by the ImageNet-pretrained SwinUnet2D variant.

a fixed train/validation/test split to ensure fair and consistent comparison with baseline methods. To prevent data leakage, all datasets were partitioned strictly at the patient level. The number of patients in the training, validation, and testing sets, respectively, is distributed as follows: CVAI (80/23/40), SegTHOR (25/5/5), Abdomen (350/100/150), and MSD10-Colon (100/26/64).

As shown in Table 2, incorporating synthetic data generated by LabelG consistently boosts segmentation accuracy across all datasets and model architectures. Compared to MAISIs, our synthetic data provides more substantial performance gains, particularly on CVAI and SegTHOR, where almost all backbones achieve the highest DSC scores under the LabelG setting. These results confirm that LabelG offers a more effective and generalizable synthetic data source for enhancing downstream medical image segmentation.

**Ablation Study of Backbone.** To assess the influence of backbone choice in LabelG, we replace our MLP-based backbone with three alternatives: SwinUnet2D (with and without ImageNet pre-training) and ResNet101. Using two 3D segmentation teachers (SwinUNet and UNETR), we generate pseudo labels for the synthetic images and compute the Sørensen–Dice coefficient (DSC) between each variant and its corresponding teacher. The average scores are summarized in Table 3.

Visual examples of the generated images and their pseudo labels are provided in Fig. 4, illustrating how different backbones affect structural consistency in the paired data.

Overall, the MLP-based backbone (backbone-LabelG) achieves the strongest alignment with teacher models, while the ImageNet-pretrained SwinUnet2D variant suffers from domain shift and performs notably worse.

Table 3: Ablation study comparing different backbones in LabelG. Dice scores are computed with respect to pseudo labels from SwinUNet and UNETR.

| Backbone Variant | SwinUNet (DSC) | UNETR (DSC) |
|---|---|---|
| backbone-LabelG (MLP) | **0.83** | **0.81** |
| backbone-SwinUnet2D_w ckpt | 0.75 | 0.71 |
| backbone-SwinUnet2D_w/o ckpt | 0.80 | 0.82 |
| backbone-ResNet101 | 0.61 | 0.66 |

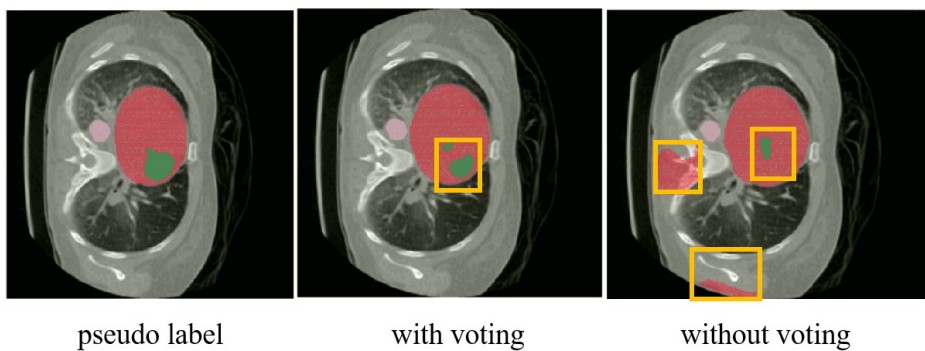

pseudo label          with voting          without voting

Figure 5: Qualitative comparison of the proposed split-and-vote mechanism on a CVAI case. Our voting-based aggregation (middle) suppresses these unstable predictions and yields a cleaner mask that better aligns with the pseudo label (left).

**Ablation on the Voting Mechanism.** We examine the role of the voting mechanism in our split-MLP design. Due to memory limitations, this comparison is conducted on a reduced subset with identical training settings. We compare a single-path model that directly processes all concatenated channels, against our split-and-vote architecture.

As shown in Fig. 5, removing voting results in unstable segmentation behavior, including spurious activations and boundary inaccuracies (yellow boxes). The voting-based aggregation suppresses these artifacts and produces cleaner masks that better match the pseudo labels from the pre-trained SwinUNet.

## 6. Conclusion

We introduced LabelG, a lightweight module for generating paired CT images and segmentation masks using a pre-trained 3D diffusion foundation model. By decoding multi-scale latent features through a split-MLP architecture and aggregating predictions via voting, LabelG can produce anatomically coherent synthetic pairs in a single sampling pass, without requiring ground-truth masks or text prompts at inference time. Experiments on four CT datasets show that the resulting synthetic pairs are of high visual fidelity and can improve segmentation performance when used to augment limited real data.

Several limitations remain. While LabelG consistently benefits organ-level segmentation, the gains for tumor segmentation are less pronounced, likely due to high morphological variability, limited appearance consistency, and stronger sensitivity to label fidelity. Addressing these challenges may require tumor-aware conditioning strategies and diversity-aware or uncertainty-guided synthesis. Our evaluation relies primarily on image-level similarity metrics; incorporating structure-aware criteria and additional expert assessment will be important for more clinically meaningful validation. Furthermore, reducing computational cost—through model compression or architectural optimization—will be essential for enabling broader use in resource-constrained settings.

## Acknowledgments

This work was supported in part by National Science and Technology Council, Taiwan under grant NSTC 113-2221-E-007-104-MY3. This research was made possible by the academic resources and advanced infrastructure provided by the National Center for High-Performance Computing, National Institutes of Applied Research (NIAR), Taiwan. We would also like to express our sincere gratitude to the medical imaging and radiology teams of the Smart Health Technology Research and Development Center at National Taiwan University Hospital (NTUH) for their assistance in data curation, annotation, and quality control.

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
