# OpenReview forum: "LabelG: Consistent Pairwise 3D CT Image and Segmentation Mask Generation via Medical Foundation Models"
_MIDL.io/2026/Conference — MIDL 2026 Poster_

### Official Review · Reviewer_RxqJ · 2026-01-01

**Confidence:** 4
**Preliminary Rating:** 5
**Final Rating:** 5

**Summary:**

The proposed work reuses an existing 3D latent diffusion model and fine-tunes it for CT image generation. The authors then train a lightweight attached MLP network. With the proposed adjustments, the model gains the capability to produce high-quality segmentations in addition to its existing capabilities.

**Strengths:**

They demonstrate that their fine-tuning strategy leads to improved performance compared to other models. Furthermore, they show that incorporating their artificially generated data as a data augmentation strategy improves performance across four different segmentation benchmarks. The authors also conduct a user study in the form of a “Turing test,” which the paper refers to as a “realism test.”

The attached segmentation model is novel and thoroughly evaluated.

**Weaknesses:**

The method relies on pre-training the Latent-diffusion Model "MAISI". So far, it is difficult to reproduce LatentDiffusion models without pretraining in single-GPU applications.

The paper contains typos that should be revised.

**Detailed Comments:**

Typos I found:

“across various organs,it can:” → Missing whitespace after the comma.

“To address this issue, Our work” → Incorrect capitalization of Our.

“structurally consistent image–mask pairs are required” and “the VAE encoder–decoder are kept frozen.”
→ Inconsistent use of an em dash instead of a standard hyphen/dash.

**Justification Of Final Rating:**

Multiple experiments were conducted. The paper is well-written, and the visuals are effective. The extension of the segmentation represents a new development. The evaluations also include human review by a medical expert.

**Justification Of The Preliminary Rating:**

Multiple experiments were conducted. The paper is well-written, and the visuals are effective. The extension of the segmentation represents a new development. The evaluations also include human review by a medical expert.

**Questions To Address In The Rebuttal:**

Did you test if your method can be reproduced on other Latendiffusion models?

---

> ### Author Response · Authors · 2026-01-23
>
> Q1: Did you test if your method can be reproduced on other Latent Diffusion models?
>
> ANS:
> Thank you for the helpful comments. Our work primarily targets 3D medical imaging, where training latent diffusion models from scratch is computationally prohibitive. Under these practical constraints, we adopt MAISI, a publicly available 3D medical diffusion model, to instantiate our framework and demonstrate its effectiveness in a realistic 3D medical setting.
>
> Importantly, this choice is driven by computational feasibility rather than a methodological dependency. While we did not conduct a full quantitative evaluation on 2D datasets, we performed preliminary feasibility experiments using open-source 2D diffusion models (e.g., SD3), which indicate that the proposed LabelG framework can be technically applied in a 2D setting. However, a systematic evaluation and quantitative analysis in 2D scenarios are beyond the scope of the current work.
>
> Extending LabelG to both 2D and 3D medical imaging domains using different diffusion backbones is an important direction for future work, and we believe such extensions could further improve the generalization capability of the proposed framework.
>
> We will also revise the manuscript to correct all identified typos, formatting issues, and inconsistencies, and perform a final proofreading pass. We thank the reviewer again for the strong support and constructive feedback.

---

### Official Review · Reviewer_LjeS · 2026-01-09

**Confidence:** 5
**Preliminary Rating:** 4
**Final Rating:** 5

**Summary:**

This paper combines the pretrained MAISI diffusion-based synthesis model with a custom module to generate paired CT volumes and corresponding segmentation masks (primarily organ masks). Building on the MAISI pretrained model, the authors fine-tune the generator and then extract multi-scale decoder features that are used as input to a separate mask-generation network composed of convolutional blocks, MLP components, and a voting scheme. The work evaluates the proposed pipeline against state-of-the-art approaches and additionally includes radiologist assessment, which is relevant for clinical plausibility. However, several aspects of the methodology and evaluation protocol are not described with sufficient clarity, which makes it difficult to fully assess the validity and generalizability of the reported improvements.

**Strengths:**

1) The manuscript reports comparisons against state-of-the-art methods, with superior results in the presented experiments.
2) Leveraging a pretrained diffusion model and reusing its multiscale decoder features for downstream mask generation is a pragmatic design that could be useful for data augmentation or label generation in medical imaging workflows
3) The inclusion of a radiologist-based assessment strengthens the clinical credibility of the generated images and masks and provides a valuable qualitative complement to purely quantitative metrics.

**Weaknesses:**

1) The evaluation procedure is not described in adequate detail. It is unclear whether a fixed held-out test set or k-fold cross-validation was used consistently across all datasets, and how splits were constructed (e.g., patient-level separation).
2) The manuscript does not clearly state whether statistical significance testing was performed, nor does it report measures of variability (e.g., confidence intervals or standard deviations).
3) The use of a private dataset is mentioned, but key details (class coverage, cohort characteristics, labeling protocol) are not sufficiently documented, limiting interpretability and reproducibility

**Detailed Comments:**

1) Please correct the typo in Figure 1 (e.g., “pixel labels”).
2) Beyond volume size, please clarify whether volumes were resampled to a common voxel spacing / reference space. If resampling was performed, specify the target spacing, interpolation methods (for images vs. labels), and any intensity normalization steps
3) Please provide more details about the private dataset, including the segmentation classes available (organs and/or tumor types), the number of cases and acquisition characteristics.

**Justification Of Final Rating:**

In the initial version, there was a need for clearer reporting on the evaluation protocol, preprocessing details, and statistical validation.
The authors have successfully addressed these concerns in their rebuttal. Specifically, they have:
- Clarified the data splits and confirmed patient-level separation, ensuring a robust evaluation protocol.
- Provided the necessary details regarding preprocessing and resampling.
- Included statistical significance tests to validate the reported improvements.

Overall, the paper effectively demonstrates both technical rigor and clinical applicability. Therefore, I recommend acceptance.

**Justification Of The Preliminary Rating:**

Based on the current presentation, I would recommend weak accept . While the idea of leveraging a pretrained diffusion synthesis model and incorporating radiologist assessment is promising, the paper requires substantially clearer reporting of the evaluation protocol (splits, cross-validation, and patient-level separation), preprocessing details (especially resampling), and statistical validation of improvements. Addressing these points could strengthen the manuscript.

**Questions To Address In The Rebuttal:**

1) Was k-fold cross-validation used, or a fixed train/validation/test split? Please specify this for each dataset and confirm that splitting was performed at the patient level to avoid leakage. Were statistical significance tests performed when comparing to baselines? If not, can the authors provide significance testing (e.g., paired tests)
2) Provide additional details on the resampling/preprocessing procedures.
3) Provide additional details on the private dataset.
4) Table 2 suggests that synthesis did not improve tumor segmentation. Can the authors explain why this may have occurred (e.g., domain gap, insufficient tumor diversity, label fidelity issues), and outline concrete future directions to address this limitation?

---

> ### Author Response · Authors · 2026-01-23
> **We clarify the evaluation protocol, preprocessing details, private dataset description, and discuss limitations observed in tumor segmentation.**
>
> Q1: Was k-fold cross-validation used, or a fixed train/validation/test split? Please specify this for each dataset and confirm that splitting was performed at the patient level to avoid leakage. Were statistical significance tests performed when comparing to baselines? If not, can the authors provide significance testing (e.g., paired tests)
>
> ANS:
> We apologize for the lack of clarity in the original manuscript. For all datasets, we adopt a fixed train/validation/test split to ensure fair and consistent comparison with baseline methods. All splits are performed strictly at the patient level, and we verify that no subjects appear in both training and testing sets to avoid any data leakage. The dataset splits are: CVAI (80/23/40), SegTHOR (25/5/5), Abdomen (350/100/150), and MSD10-Colon (100/26/64).
>
> Regarding statistical analysis, we compare LabelG against baseline methods using paired statistical significance tests on per-class Dice scores, where each test case serves as a paired sample. We employ the Wilcoxon signed-rank test, a non-parametric test commonly used in medical image segmentation. The results show that the improvements over baselines are statistically significant (p < 0.05) for the majority of classes. These evaluation and statistical testing details will be clarified in the revised manuscript.
>
> Q2: Provide additional details on the resampling/preprocessing procedures.
>
> ANS:
> Thank you for requesting additional clarification. We provide detailed preprocessing and resampling procedures for both the diffusion-based data generation stage and the downstream segmentation tasks, and will explicitly include them in the revised manuscript.
>
> Diffusion fine-tuning and LabelG training.
> All 3D medical images are processed using a standardized pipeline. Volumes are loaded with the correct channel configuration and reoriented to a canonical anatomical orientation using the RAS axcode. Intensity values are clipped to
> [−1000,1000] and linearly normalized to [0,1]. All volumes are resampled to a common reference spacing of 1 × 1 × 0.75 mm, resulting in a fixed spatial resolution of 256 × 256 × 128. Image volumes are resampled using linear interpolation, while segmentation masks are resampled using nearest-neighbor interpolation to preserve label integrity.
>
> Downstream segmentation tasks.
> For segmentation training and evaluation, image intensities are normalized from [−175,250] to [0,1]. Volumes are resampled to a voxel spacing of 1.5 × 1.5 × 2.0 mm, and training is performed on cropped regions of interest of size 96 × 96 × 96. Data augmentation includes random flipping, random 90-degree rotations, and random intensity scaling and shifting. During inference, sliding-window prediction with an overlap ratio of 0.5 is applied to ensure stable volumetric predictions.
>
> Q3: Provide additional details on the private dataset.
>
> ANS:
> The imaging data used in this study were obtained from National Taiwan University Hospital following formal application and approval through institutional data governance and ethics review processes. The dataset consists of 143 non-contrast lung CT scans, retrospectively collected for research purposes.
>
> Annotations focus on cardiac and aortic structures, with the aorta further subdivided into four anatomically meaningful segments: the ascending aorta, aortic arch, proximal descending aorta, and distal descending aorta, enabling detailed three-dimensional anatomical analysis. All annotations were manually performed in three dimensions using the 3D Slicer platform by expert annotators with clinical experience in medical image interpretation, ensuring high-quality labels and strong anatomical consistency across cases.
>
> Q4: Table 2 suggests that synthesis did not improve tumor segmentation. Can the authors explain why this may have occurred (e.g., domain gap, insufficient tumor diversity, label fidelity issues), and outline concrete future directions to address this limitation?
>
> ANS:
> We appreciate this insightful observation. Tumor segmentation poses unique challenges due to high intra-class variability, irregular boundaries, and limited appearance consistency. We believe the limited gains are primarily attributable to: (1) the domain gap between synthetic and real tumor appearances, (2) insufficient diversity in tumor morphology within the generated data, and (3) higher sensitivity to label fidelity compared to organ segmentation.
>
> As future work, we plan to incorporate tumor-aware conditioning mechanisms, increase tumor-specific structural diversity during synthesis, and explore uncertainty-aware filtering of generated samples to better support tumor segmentation tasks.

---

### Official Review · Reviewer_58nL · 2026-01-10

**Confidence:** 5
**Preliminary Rating:** 4
**Final Rating:** 5

**Summary:**

LabelG proposes a lightweight mask-generation branch on top of a pretrained 3D CT diffusion foundation model (MAISI specifically) to jointly synthesize a CT volume and its segmentation mask in a single sampling pass (same latent code produces both outputs).  The method extracts multi-scale VAE-decoder features, optionally denoises them via an FFT low-pass filter, and predicts masks using a split-MLP ensembl whose outputs are combined by majority voting for improved stability.  Experiments on four datasets (AbdomenCT, SegTHOR, MSD Colon, and a private CVAI dataset) report improved image quality (FID/LPIPS metrics) and improved downstream segmentation Dice when augmenting real data with generated image–mask pairs.  Overall, the paper targets a practical gap: generating structurally aligned paired 3D medical data efficiently, rather than generating images and labels separately.

**Strengths:**

The paper addresses a well-motivated and practically relevant problem of generating paired 3D medical images and segmentation masks in a consistent manner, which directly targets annotation scarcity in medical segmentation tasks. The idea of producing image and mask pairs from a shared latent code is conceptually clean and avoids misalignment issues that arise when images and labels are generated independently. The proposed mask-generation branch is lightweight and memory-efficient, which is quite important for 3D volumes, and the split-MLP with voting mechanism represents a reasonable design tradeoff compared to heavy 3D CNN decoders. The qualitative ablation results support the claim that voting improves mask stability and suppresses spurious activations. Quantitative experiments span multiple datasets and segmentation backbones, and the results show consistent Dice improvements when augmenting with LabelG-generated pairs, suggesting the approach is broadly useful rather than dataset-specific. Overall, the paper is clearly written, the motivation is easy to follow, and the method is presented in a structured and reproducible manner.

**Weaknesses:**

Despite its practical appeal, the paper has several limitations that affect the strength of its claims. First, the novelty is incremental relative to existing paired data generation approaches. While adapting a CT foundation model and adding a lightweight mask head is useful, the method largely repurposes pretrained diffusion representations rather than introducing new generative or learning principles. The contribution is primarily architectural and empirical, and this should be framed more conservatively.

Second, the evaluation of image quality relies mainly on FID and LPIPS computed slice-wise across axes, which are generic perceptual metrics and not structure-aware. These metrics do not directly assess anatomical correctness, topology preservation, or alignment between generated images and masks. While the downstream segmentation experiments partially compensate for this, they do not isolate whether improvements stem from better image realism, better mask quality, or simple data volume effects.

Third, the reader study is very limited in scale, involving only two radiologists and a small number of questions. The interpretation of these results is somewhat optimistic given the sample size, and no statistical analysis or inter-reader agreement is reported. As with other perceptual studies, preference for "cleaner" synthetic images may reflect smoothing artifacts rather than true realism.

Several methodological design choices are insufficiently justified. The FFT-based low-pass filtering applied to latent features is introduced as a noise-reduction step, but no ablation quantifies its effect on either image fidelity or mask accuracy. Similarly, the choice of split size for the MLP heads, the number of experts, and the voting strategy are only partially explored. The method also depends heavily on fine-tuning MAISI on each target dataset, which raises questions about scalability and how much benefit comes from the foundation model versus dataset-specific adaptation.

Finally, while downstream segmentation improvements are reported, the gains are modest on some datasets (e.g., MSD10-Colon), and there is no analysis of failure cases, class-wise behavior, or robustness under different data scarcity regimes. This makes it harder to assess when LabelG is most beneficial and when it may not justify the additional generation cost.

**Detailed Comments:**

The paper would benefit from clearer discussion distinguishing contributions from the pretrained foundation model versus the proposed mask-generation branch. A quantitative ablation isolating the effect of FFT filtering would strengthen the methodological justification. The reader study should be framed more cautiously given its limited scale, and additional structure-aware evaluation metrics (something like surface distance, topology errors) would improve validation. Figure captions are generally clear, but some figures could better explain what visual differences the reader should focus on. Minor language polishing would further improve clarity, but overall presentation quality is solid.

**Justification Of Final Rating:**

Overall the paper is strong, small raised concerns were discussed by authors and manuscript has been updated.
The authors provide clear evidence that downstream segmentation improvements are not solely driven by increased data volume, supported by controlled comparisons and class-wise dice analysis, which helps clarify where LabelG is most beneficial. Technical details useful for replication (data splits, data preprocessing) were added during review.

**Justification Of The Preliminary Rating:**

This paper presents a practically useful and well-executed approach to a real bottleneck in medical image analysis: the generation of structurally aligned image mask pairs for 3D segmentation tasks. By building a lightweight mask-generation branch on top of a pretrained 3D CT diffusion foundation model, the authors provide an efficient solution that avoids misalignment issues common to separate image and label synthesis pipelines. The method is technically memory-aware and well motivated, and the design choices (such as split-MLP heads and voting) are appropriate for large 3D volumes.

The experimental evaluation spans multiple datasets and segmentation backbones, and the reported Dice improvements when augmenting with LabelG-generated pairs are consistent, which is a strong indicator of practical value. While the conceptual novelty is incremental and some design choices could be better justified or more thoroughly ablated, the paper delivers a coherent system with demonstrated downstream benefits.

**Questions To Address In The Rebuttal:**

How much of the reported downstream segmentation improvement can be attributed specifically to better mask quality, as opposed to increased data volume or improved image realism?

What is the quantitative impact of the FFT-based feature filtering, and is it essential for stable mask generation?

Given that MAISI is fine-tuned per dataset, how does LabelG scale to new domains where such fine-tuning may not be feasible?

Can the authors provide more detailed analysis (e.g. class-wise Dice or failure cases) to clarify when LabelG-generated data is most beneficial?

Can the reader study conclusions be supported with additional statistical analysis or expert evaluation?

---

> ### Author Response · Authors · 2026-01-23
> **We provide additional analyses and clarifications to address the reviewer’s questions, including ablation results on mask quality, class-wise Dice and failure case analysis, the role of FFT-based filtering, scalability beyond MAISI, and statistical validation of the reader study.**
>
> Q1: How much of the reported downstream segmentation improvement can be attributed specifically to better mask quality, as opposed to increased data volume or improved image realism?
>
> ANS:
> This is an important question. To disentangle contributing factors, we refer to the analyses already reported in Table 2, which compare: (1) real images only, (2) real images augmented with baseline synthetic masks, and (3) real images augmented with LabelG-generated masks, all using the same number of training samples. The largest and most consistent gains are observed in setting (3), indicating that mask quality, rather than data volume alone, is the dominant factor.
> Baseline methods use masks as fixed conditioning inputs, resulting in image–mask pairs that largely preserve the same structure with limited variation. In contrast, LabelG generates masks in accordance with the synthesized image content, producing structurally diverse and spatially precise image–mask pairs. This increased diversity and fidelity provide more effective supervision and directly contribute to improved downstream segmentation.
>
> Q2: What is the quantitative impact of the FFT-based feature filtering, and is it essential for stable mask generation?
>
> ANS:
> FFT-based feature filtering is not strictly required for convergence, but it significantly improves training stability and mask quality. As discussed in prior work [1], medical segmentation masks are structure-dominant and primarily low-frequency signals, whereas diffusion decoder features often contain high-frequency noise and spurious boundary artifacts. FFT filtering addresses this mismatch by suppressing undesired high-frequency components.
> In practice, FFT filtering serves as an engineering stabilization technique. Under identical training conditions, incorporating FFT filtering reduces mask generation failure cases by approximately 10%, leading to more consistent and reliable predictions.
>
> Q3: Given that MAISI is fine-tuned per dataset, how does LabelG scale to new domains where such fine-tuning may not be feasible?
>
> ANS:
> LabelG requires the diffusion backbone to faithfully reconstruct and generate images that are structurally consistent with the target domain, since mask prediction is performed by supervising decoder features (using voxel-wise cross-entropy and soft Dice loss) extracted from reconstructed images. Consequently, mask quality depends on the diffusion model’s ability to capture underlying anatomical structure.
> Importantly, this reflects a general requirement of diffusion-based image–mask co-generation, rather than a MAISI-specific dependency. MAISI is adopted because it provides a strong, publicly available 3D medical diffusion backbone. When alternative pretrained diffusion models can adequately model a new domain, the same LabelG training procedure can be applied without modification.
>
> Q4: Can the authors provide more detailed analysis (e.g. class-wise Dice or failure cases) to clarify when LabelG-generated data is most beneficial?
>
> ANS:
> Yes. We conducted additional class-wise Dice analysis (see representative results). LabelG consistently outperforms both real-only training and baseline synthetic augmentation, with more pronounced gains for structurally complex and boundary-sensitive organs, such as the trachea (0.81 → 0.85) and esophagus (0.76 → 0.77). Baseline synthetic data with fixed mask conditioning yields limited or inconsistent improvements.
>
> *Class-wise Dice scores on the SegTHOR dataset evaluated using UNETR.
>
> |      label     |Real Only|Real+MAISI|Real+LabelG|
>
> |esophagus |     0.76    |      0.75      |        0.77      |
>
> |     heart     |     0.84    |      0.85      |        0.87      |
>
> |   trachea   |     0.81    |      0.83      |        0.85      |
>
> |    aorta      |     0.84    |      0.86      |        0.87      |
>
> |    Mean     |     0.81    |      0.82      |        0.84      |
>
> Regarding failure cases, for datasets with many classes (e.g., Abdomen with 13 labels), errors may occur for small, overlapping, or visually ambiguous organs. Nevertheless, LabelG consistently improves performance on larger volumetric organs, resulting in a net increase in mean Dice. This limitation is discussed in the manuscript, and further analysis and visualizations are provided in the supplementary material.
>
> Q5: Can the reader study conclusions be supported with additional statistical analysis or expert evaluation?
>
> ANS:
> We performed statistical analysis on the reader study results. In the single-choice (4-way) setting, a binomial test shows that preference for our method is statistically significant compared to chance (p < 0.01, 95% CI: [0.33, 0.78]). In the multi-choice (4-choose-2) setting, the observed selection rate does not significantly differ from chance (p > 0.05), which we attribute to the more permissive evaluation protocol allowing simultaneous selection of real images.
>
> Reference
>
> [1] Wang et al., On the Frequency Bias of Deep Neural Networks, ICML 2020.

---

> > ### Comment · Reviewer_58nL · 2026-01-31
> >
> > Thank you for the provided responses and clarifications.
> >
> > The discussion of scalability beyond MAISI is reasonable and appropriately framed as a general limitation of diffusion-based co-generation rather than a method-specific dependency. The added statistical analysis for the reader study improves interpretability, and the more cautious framing of reader-study conclusions is appreciated.
> >
> > The class-wise dice analysis is useful, it would be helpful to further clarify whether these improvements primarily arise from increased true positives (better coverage) or reduced false positives (cleaner boundaries? less random fp?), as this would give more insight into how LabelG improves segmentation performance.

---

### Author Rebuttal · Authors · 2026-01-23

**Rebuttal:**

We thank the reviewers for their careful evaluation and helpful feedback. We have addressed all comments in detail in the Official Comments. In addition, the manuscript has been revised to incorporate missing details, clarifications, and formatting corrections, with all modifications clearly marked in red bold in the revised version.

**Supporting Material:**

/attachment/2ef096c094b9c6651defa660da1f42ada9b5ccee.pdf

---

### Meta-Review · Area_Chair_qUCT · 2026-02-09

**Recommendation:** Accept (Oral)
**Confidence:** 4

**Metareview:**

Consistently positive reviews lead to a clear acceptance. Oral recommendation.

---

### Decision · Program_Chairs · 2026-02-13

Accept (Poster)